# Captopril Alleviates Chondrocyte Senescence in DOCA-Salt Hypertensive Rats Associated with Gut Microbiome Alteration

**DOI:** 10.3390/cells11193173

**Published:** 2022-10-10

**Authors:** Lok Chun Chan, Yuqi Zhang, Xiaoqing Kuang, Mohamad Koohi-Moghadam, Haicui Wu, Theo Yu Chung Lam, Jiachi Chiou, Chunyi Wen

**Affiliations:** 1Department of Biomedical Engineering, The Hong Kong Polytechnic University, Hung Hom, Kowloon, Hong Kong 999077, China; 2Research Institute for Smart Ageing, The Hong Kong Polytechnic University, Hung Hom, Kowloon, Hong Kong 999077, China; 3Department of Applied Biology and Chemical Technology, The Hong Kong Polytechnic University, Hung Hom, Kowloon, Hong Kong 999077, China; 4Faculty of Dentistry, The University of Hong Kong, Pokfulam, Hong Kong 999077, China; 5Department of Civil and Environmental Engineering, Imperial College London, London SW7 2AZ, UK

**Keywords:** gut microbiota, chondrocyte senescence, hypertension, captopril

## Abstract

Gut microbiota is the key controller of healthy aging. Hypertension and osteoarthritis (OA) are two frequently co-existing age-related pathologies in older adults. Both are associated with gut microbiota dysbiosis. Hereby, we explore gut microbiome alteration in the Deoxycorticosterone acetate (DOCA)-induced hypertensive rat model. Captopril, an anti-hypertensive medicine, was chosen to attenuate joint damage. Knee joints were harvested for radiological and histological examination; meanwhile, fecal samples were collected for 16S rRNA and shotgun sequencing. The 16S rRNA data was annotated using Qiime 2 v2019.10, while metagenomic data was functionally profiled with HUMAnN 2.0 database. Differential abundance analyses were adopted to identify the significant bacterial genera and pathways from the gut microbiota. DOCA-induced hypertension induced p16INK4a+ senescent cells (SnCs) accumulation not only in the aorta and kidney (*p* < 0.05) but also knee joint, which contributed to articular cartilage degradation and subchondral bone disturbance. Captopril removed the p16INK4a + SnCs from different organs, partially lowered blood pressure, and mitigated cartilage damage. Meanwhile, these alterations were found to associate with the reduction of *Escherichia-Shigella* levels in the gut microbiome. As such, gut microbiota dysbiosis might emerge as a metabolic link in chondrocyte senescence induced by DOCA-triggered hypertension. The underlying molecular mechanism warrants further investigation.

## 1. Introduction

Gut microbiota is the key controller of healthy aging in older adults [1]. Mounting evidence suggests that the gut microbiome signature reflects healthy aging and predicts survival in humans [2]. Moreover, alteration in the gut microbiota emerges as a shared pathomechanism underlying a variety of age-related pathologies such as hypertension [3] and OA [4].

Vascular attrition leads the way to systemic aging, which precedes the emergence of cellular aging’s hallmarks, such as cellular senescence [5]. Hypertension, one of the most common types of vascular attrition, induces endothelial senescence and gives rise to age-related vascular pathologies [6,7]. It has also been shown to induce somatic cell senescence in a DOCA-induced hypertensive rat model, while the anti-hypertensive drugs could ablate p16^INK4a^-positive senescent cells from the body [6]. Importantly, p16^INK4a^-positive senescent cell accumulation was corroborated to engender joint damage [8] and bone loss [9]. Intra-articular senescent chondrocytes impair the cartilage regeneration capacity of mesenchymal stem cells [10]. Captopril, a widely used anti-hypertensive drug targeting angiotensin-converting enzyme to suppress the production of the vasoconstrictor-angiotensin II, can extend the lifespan of *Caenorhabditis Elegans* [11]. Our recent work demonstrated the association between DOCA-induced hypertension and gut microbiota in the rat model; meanwhile, the anti-hypertensive effect of captopril was also linked with gut microbiome alteration [12].

So far, there are three unanswered questions. The first question is whether hypertension induces cellular senescence in bone and joints, and the second is whether captopril can remove cellular senescence in bone and joints in the DOCA-induced hypertensive rat model. The last one is whether the gut microbiota plays a role in the emergence and removal of cellular senescence in bone and joints in the DOCA-induced hypertensive rat model.

To fill this gap, in this study, through the establishment of a DOCA systemic aging model, we first elucidated the effect of hypertension on the articular cartilage and subchondral bone. Later, with the administration of the anti-hypertensive drug captopril, we demonstrated the removal of senescence in the joint tissue and the partial restoration effect of the bone. In particular, this study focuses on the therapeutic effect of the drug on suppressing the established hypertensive condition rather than its preventive function for hypertension. The composition and correlation structure of the gut microbiome was subsequently examined under hypertensive and normotensive conditions, from which key bacterial genera were further identified that potentially associate with senescence change under the influence of differential blood pressure.

## 2. Materials and Methods

The study adopted a sample size of 8 for each group (i.e., a total sample size of 24, assuming equal group sizes). We first referred to the mean and standard deviation of blood pressure data of the control and DOCA-treated groups reported by Bae et al. [13]. Later, with the online statistical tool, Statulator [14], we calculated the required sample size to be at least 5 per group in order to achieve a power of 80% and a level of significance of 5% (two-sided) for detecting a true difference in means between the test and the reference group. Based on the estimation, the sample size was further increased to 8 per group to ensure sufficient statistical power.

A total of 24 male Sprague Dawley (SD) rats weighing 180–200 g (around 6 weeks old) were used in this study. The rats were randomly divided into the control, DOCA-induced hypertensive, and captopril-treated groups, each comprising 8 rats. The hypertensive and captopril-treated (i.e., DOCA + Captopril) groups received a subcutaneous injection of DOCA (20 mg/kg bw) twice a week, in combination with 1.0% NaCl and 0.2% KCl in the drinking water to induce hypertension. While rats in the control group were injected with saline, no saltwater was supplied. This treatment lasted for 14 weeks. The DOCA + Captopril group was administered with captopril (50 mg/kg) by oral gavage daily since week 9. Fecal samples were collected at the end of week 14. The body weight was measured every week, while the blood pressure was recorded every two weeks by the tail-cuff method using a BP-2000 Blood Pressure Analysis System (Visitech System, Inc., Apex, NC, USA). The aorta, kidney, and liver samples were collected for routine histology and immunostaining to examine the accumulation of p16^INK4a^ + senescent cells.

### 2.1. Micro-CT Analysis

Micro-CT images of the knee joint were obtained. The images were subsequently analyzed using DataViewer (version 1.4.4.0, SKYSCAN) with the grey level threshold for binarization set to 90. The bone volume fraction (BV/TV), trabecular separation (Tb.Sp), trabecular number (Tb.N), and trabecular thickness were then measured with the in-built tools of the software. Later, radiomic analysis was conducted using the PyRadiomics v3.0.1 package from Python 3.7 [15]. The first order, Gray Level Co-occurence Matrix (GLCM), Gray Level Run Length Matrix (GLRLM), Gray Level Size Zone Matrix (GLSZM), Neighbouring Gray Tone Difference Matrix (NGTDM), and Gray Level Dependence Matrix (GLDM) features were calculated from the 3-dimensional images using a bin width of 55, forming a total feature set of 1459 variables. Finally, top-50 radiomic features were selected using the chi-square test, and their mean values were plotted as heatmaps.

### 2.2. Histology

The samples harvested from the rat were first fixed with 4% paraformaldehyde for 24 h, followed by decalcification of the bones with 10% Ethylenediaminetetraacetic acid. Then, the samples were embedded with wax after the tissue processing procedure. A microtome was used to cut the sample into 5μm sections, and immunohistochemical staining was performed to detect specific proteins in the samples. After dewaxing and antigen retrieval, we used horse serum for blocking. The samples were then incubated with the primary antibody overnight at 4 °C, where the primary antibodies used were p16 (1:500; Abcam, ab54210), p53, and MMP13 antibody (1:500; Abcam, ab39012). For 3,3’-Diaminobenzidine (DAB) staining, we used a Vectastain ABC kit and a DAB peroxidase substrate kit (Vector Labs, Newark, CA, USA) to stain the targeted antigen. Then Harris hematoxylin was employed for counterstaining. We employed Safranin O/Fast green for Safranine O staining on the tibia. In each rat, 10 slices were selected for analysis on which both menisci appeared to be disconnected, and finally, 3 slices were chosen for detailed analysis. Specifically, the cells with brown nuclei were identified as p16-positive cells. All images were taken with a Nikon Eclipse 80i microscope (Nikon, Tokyo, Japan). The data were then compared using one-way ANOVA with posthoc Tukey HSD test in SPSS statistical analysis software (IBM Software, Armonk, NY, USA).

### 2.3. Gut Microbiota Analysis

Stool samples were collected sterilely from the rectum at week 14 and stored at −80 °C. The 16S rRNA amplicon sequencing targeted V3V4 regions, and shotgun metagenomic sequencing was performed using Illumina HiSeq (Illumina, Inc., San Diego, CA, USA) PE250/PE300 sequencer (300–500 bp paired-end reads). Subsequently, the 16S rRNA data were processed and annotated using Qiime 2 v2019.10 [2]. The raw paired-end sequences of the metagenome data were first denoised, merged, then functionally profiled using HUMAnN 2.0 [5] and PICRUSt to reconstruct species-level microbial metabolic pathways as well as molecular functions of microbiota. A total of 2659 pathways were successfully profiled in the end.

A genus-level TAXA plot was used to visualize the top 20 abundant genera identified from gut bacterial 16S rRNA gene amplicon sequencing for each group. Later, the richness and Shannon diversity measures were calculated for each sample. A cladogram was plotted to demonstrate the linear discriminant effect size of the significant gut genera (*p* < 0.05). Alterations in the macroscopic correlation structure among genera in the gut microbiome under hypertension and captopril treatment were assessed by constructing Spearman’s correlation networks for each experimental group. The network was constructed by only considering the magnitude of associations between the genera. As a result, the network edges were defined as the absolute values of the correlation coefficients, and only correlations with *p* < 0.05 were retained. The network nodes were subsequently clustered by the spectral clustering algorithm using Scikit-Learn 1.0.2 to visualize emerging structures. Finally, the global network parameters, including centrality, density, heterogeneity, clustering coefficient, and Shannon entropy, were calculated with 1000 iterations of bootstrap sampling using the igraph version 1.2.7 package of R 4.0. 

Genera with significant differential abundance across the experimental groups were identified using DESeq2 [16] out of 200 genera profiled from the 16S rRNA sequencing data. The between-group fold changes were plotted, and the top 2 principal components of the selected genera were visualized to illustrate the distribution of the differential abundance across different experimental groups. Additionally, the bacterial pathways from metagenomic data showing significant differential expression under DOCA-induced hypertension and captopril treatment were selected using the EdgeR algorithm [17]. The groupwise fold-change of expression level was plotted in log-scale, and the first two principal components of the selected pathways were also plotted.

### 2.4. Multi-Omic Correlation Network

To unravel the correlation structure linking multiple omics of data, including blood pressure, bone phenotypes of primary and secondary spongiosa, selected bacterial genera, and the enriched pathways under DOCA-induced hypertension and the captopril administration, we constructed a multi-omic correlation network by integrating samples from all three experimental groups using igraph version 1.2.7 package of R. All data were rescaled to a range of 0 to 1. Only Spearman’s correlations with Benjamini and Yekutieli-adjusted *p* < 0.05 were retained and network edges were defined as the absolute values of the correlation coefficients.

### 2.5. Statistical Analysis

Inter-group comparisons of the p16 and MMP13 staining were compared using one-way ANOVA with posthoc Tukey HSD test in SPSS statistical analysis software (IBM Software, Armonk, NY, USA). A permutational multivariate analysis of variance (PERMANOVA) and pairwise Adonis analyses were conducted to test for the difference in composition of the gut microbiota among the control, hypertensive, and captopril-treated groups. The group means of bone phenotypes, gut flora richness, gut microbiome correlation network parameters, genus abundance, and bacterial pathway expression level were compared using one-way ANOVA with Tukey HSD posthoc analysis using R 4.0.3.

## 3. Results

### 3.1. Senescent Cells Accumulation with the Onset of Metabolic OA after DOCA Induction

DOCA induced the p16^INK4a^-positive senescent cell accumulation in both aorta and the kidney, which ultimately contributed to elevated blood pressure (Figure 1). However, it did not alter the body weight over time (Figure A1).

Meanwhile, DOCA also triggered the p16^INK4a^-positive senescent cells accumulation in different knee joint tissues such as articular cartilage, synovium, meniscus, and subchondral bone (Figure 2A–F), as well as the enhancement of p53 expression in the articular cartilage (Figure 2I,J). As one of the senescence-associated secretory phenotypes (SASP), the expression of MMP-13 is upregulated in articular cartilage and synovium (Figure 2K–N).

As a consequence, articular cartilage appeared thinner with loss of proteoglycan in the DOCA group than in the control group (Figure A2). The increase in p16 staining has also been observed in the subchondral bone (Figure 3A,B), leading to significant bone loss (Figure 3C–H). Similar findings were also found in the primary spongiosa beneath the growth plate (Figure A3).

### 3.2. Captopril Reduced Senescent Cells and Mitigated Joint Deterioration

Captopril reduced the number of senescence cells in the aorta and kidney and lowered blood pressure (Figure 1B–D). It could also decrease senescent cell accumulation and MMP-13 expression in synovial joint tissues, ultimately preserving articular cartilage in hypertensive rats (Figure 2). In addition to articular cartilage, captopril could reverse subchondral cellular senescence (Figure 3A–C) and bone radiomics change (Figure 3F) after DOCA induction. Noteworthily, there only existed a trend for the osteo-protective effect of captopril in DOCA rats in terms of BV/TV and trabecular bone thickness, but it lacked statistical significance (Figure 3C–E). In contrast, captopril failed to restore bone mass and microstructure in primary spongiosa under the growth plate, although it could mitigate senescence cell accumulation (Figure A3).

### 3.3. Rebalancing Effect of Captopril on Gut Microbiota in DOCA-Induced Hypertensive Rats

DOCA induction altered the composition of the gut flora in association with hypertension and metabolic OA in a rat model. As shown in Figure 4A,B, the TAXA plot and cladogram display the disparity in the phylogenetic distribution of the bacterial lineages and top 20 most abundant genera between the control and DOCA groups. However, the conventional analysis of the richness of microbial composition failed to demonstrate statistical significance, although the trend was observed. Therefore, we developed a novel approach to analyze the organization of the gut microbiome in terms of its correlation structure mathematically. Intriguingly, the cluster structure of the gut microbiome in the control group was broken into a few small clusters in the hypertensive group (Figure 4D). Moreover, the organization of the gut microbiome of hypertensive rats exhibited increasing density and Shannon entropy in the DOCA group, while density and Shannon entropy decreased in the Captopril group (Figure 4E–H). Similar trends in the disparity of gut microbiome among groups have been further validated by the Permutational multivariate analysis of variance (PERMANOVA) (Table A1) and pairwise Adonis (Table A2).

Captopril did not significantly alter the overall abundance of the gut flora genera (*p* = 0.68) (Figure 4A and Table A1 and Table A2). However, it reduced the richness of the microbiome with marginal significance (*p* = 0.065). Furthermore, it significantly changed the organization of the gut microbiome in terms of density and Shannon entropy back to a level similar to the control. It is, therefore, evident that captopril might show a trend to partially restore certain microbial macroscopic structures in the gut.

### 3.4. Identification of Differentially Activated Pathways of Gut Microbiota Associated with Senescent Cells Removal Using Captopril

Among all enriched genera from the 16S rRNA sequencing, four of them, *D**esulfovibrio*, *Victivallis*, *Escherichia-Shigella,* and *Lachnospriraceae-UGC006,* were isolated. *Escherichia-Shigella* exhibited differential abundance upon the induction of hypertension by DOCA and, at the same time, showed signs of rebalancing after captopril administration (Figure 5E). Alongside, other genera, Desulfovibrio and Lachnospriraceae-UGC006, were suppressed, whereas RVictivallis was enriched under the hypertensive condition. Nonetheless, the captopril treatment showed no restoration to the abundance in any of the three genera (Figure 5C,D,F).

### 3.5. Identification of Differentially Activated Pathways of Gut Microbiota Associated with Senescent Cells Removal Using Captopril

To further elucidate the potential involvement of the bacterial pathways in the gut flora, differential expression analysis was carried out. Using the EdgeR statistical model, 32 pathways were identified to be significant among the 3 groups, where the PC plot constructed using the selected 32 pathways (Figure 6B) also shows a clear separation between the Control, DOCA, and Captopril-treated groups. With a further selection, we have identified 2 bacterial pathways (Figure 6C,D) that potentially relate to the effect of captopril on the alleviation of hypertension and joint deterioration.

### 3.6. Multi-Omic Correlation Analysis Unraveled Associations between Microbiota and Bone under the Hypertensive and Normotensive Conditions

A multi-omics correlation network was constructed (Figure 7) that integrates the previously selected gut 16S bacterial genus, the pathways identified from the gut metagenomic data, blood pressure, as well as the phenotypic data of secondary spongiosa, including bone volume fraction (BV/TV), trabecular thickness (Tb.Th), trabecular separation (Tb.Sp), and trabecular number (Tb.N).

On the macroscopic view of the networks, a large cluster forms around blood pressure involving correlations between multiple data sets. Blood pressure correlates with BV/TV. BV/TV and Tb.N of secondary spongiosa was found to associate with the Desulfovibrio. The blood pressure correlated with Lachnospiracea.UCG.006 and Victivallis. Additionally, L-valine biosynthesis (g2633), adenosine deoxyribonucleotides de novo biosynthesis (g2004), and guanosine deoxyribonucleotides de novo biosynthesis (g2063) were correlated with blood pressure.

## 4. Discussion

Articular cartilage damage is the primary concern of osteoarthritis (OA). In response to altered mechanical loading (e.g., after injury) or oxidative stress (e.g., aging), articular chondrocytes undergo premature senescence and stop dividing permanently, which provokes the onset of OA [8]. Our results demonstrate articular cartilage deterioration in a DOCA-hypertensive rat model, in which a significant elevation in the p16 and MMP13 expression was observed in the joint tissues, including cartilage, meniscus, and synovium. Overexpression of cellular senescence marker p16^Ink4a^ in chondrocytes induces cartilage degradation with two matrix remodeling enzymes, i.e., matrix metalloproteinase (MMP)-1 and −13 [18]. Moreover, very recent studies provided direct evidence to show the involvement of senescent cells in cartilage damages [8,19], where ablation of p16^Ink4a^-positive cells using a genetically modified mice model could mitigate OA [8]. The major flipside up to this moment is that the mechanism leading to age-related cartilage degradation remains unclear and early changes that predispose to chondrocyte senescence and cartilage matrix disruption are not well characterized. This finding sheds light on the current situation by the indication of hypertension as a potential risk factor triggering chondrocyte senescence. The more noteworthy experimental finding is that captopril treatment reduces the articular cartilage senescence back to the normal level. This might be attributed to the anti-hypertensive effect of captopril [20], which eventually removes the risk factor triggering senescence in the joint tissue.

Notable subchondral bone loss was observed in the DOCA-hypertensive rats; however, restoration was not exhibited in the bone volume ratio, trabecular separation, and trabecular number. Interestingly, in contrast to other bone phenotypes, the trabecular thickness in secondary spongiosa showed a slight increase after the captopril treatment. Moreover, through the comprehensive radiomic analysis of the micro-CT images of both primary and secondary spongiosa, we successfully identified the distinctive difference of the radiomic marker “fingerprint” in the hypertensive group when compared to the control and captopril-treated groups. This may imply the ability of captopril to alter the bone texture that is not obvious enough to be manifested in the above bone phenotypes. Overall, captopril poses a senolytic effect on the cartilage and partially restores the trabecular structure.

DOCA-induced hypertension disrupts the global structure of the gut microbiome, in which a higher degree of disorderedness of the correlation network leads to the dissociation of large correlation clusters in the normotensive samples into smaller distinct groups. It has been reported that hypertension is associated with the occurrence of dysbiosis through increasing lipopolysaccharide biosynthesis, steroid degradation [21], and gut permeability, which results in a shift in the overall composition of the gut flora [22]. From our experiment, the administration of captopril exerts a restoration effect on the macroscopic microbiome structure over DOCA-induced hypertension, reducing richness and correlation complexity back to the level comparable to the control. This echoes a previous study reporting that the anti-hypertensive drug reshapes the microbiome via a continuous influence on specific microbial populations, hence improving dysregulated hypertensive rats’ gut–brain axis [20].

In synchrony with our previous work [11], Escherichia-Shigella, which showed higher abundance under the DOCA-hypertensive condition, is rebalanced upon the administration of captopril. In addition to its association with hypertension, our results further demonstrated that under high blood pressure, the genus is closely correlated with senescence. Combined with findings of increased senescence in the secondary spongiosa, Escherichia-Shigella might be involved specifically in hypertension-induced joint senescence.

We also observed *Victivallis* to be over-expressed under hypertensive conditions, which failed to be rebalanced by captopril. This genus was found in our results to be correlated with the trabecular bone thickness of secondary spongiosa under both hypertensive and captopril-induced anti-hypertensive environments. Previous studies revealed that *Victivallis* is positively associated with hepatic lipid accumulation [23]. Notably, increased liver adipose fat has been reported as one of the critical risk factors for osteoarthritis [24], where it corresponds to a higher circulating level of proinflammatory factors, including IL-6 in the serum, triggering the anti-osteoblastic effect that matches the observed reduction in secondary spongiosa trabecular bone thickness. However, we speculate that the insignificant decrease in the genus’ abundance level upon the captopril treatment might contribute to the lack of notable restoration in the subchondral bone, except the alteration in radiomic markers and the increase in trabecular bone thickness which has direct correspondence to the osteoblast activity.

*Desulfovibrio* exhibited lower abundance in DOCA-induced hypertension, and captopril failed to restore it. The genus has been observed with elevated abundance in symptomatic hand OA [25]. In contrast, very recent work by Yu et al. pointed out that the higher level of *Desulfovibrionales* order, to which *Desulfovibrio* belongs, is beneficial to knee OA [26]. Experiments observed that the abundance of family *Desulfovibrionaceae* in Interleukin-1α (IL-1α) knockout mice are notably higher than in the control group [27], indicating a potential role of the bacterial order in the inflammatory response that triggers the onset of OA. The above evidence suggests bacterial participation in affecting kidney function and OA development; however, in light of the seemingly opposite role of it in symptomatic hand OA and knee OA, further work has to be carried out to characterize the relationship between the genus and different types of OA.

Other genera, including *Lachnospriraceae-UGC006*, were suppressed. *Lachnospriraceae*, the family of *Lachnospriraceae-UGC006*, demonstrated a lower abundance level in the symptomatic hand OA cases [25]. Given the associations, it is speculated that the hypertensive environment leads to a decrease in the genus, which could act against the deterioration of the knee joint. Particularly, all three genera were not rebalanced by captopril treatment.

Our results have demonstrated the potential of the anti-hypertensive drug captopril in alleviating chondrocyte senescence and partially reversing joint aging under the DOCA-induced hypertensive condition, where *Escherichia-Shigella* in the gut microbiota might play a role in the restoration mechanism. However, our study did not consider the possible preventive effect of the drug on joint deterioration in normotensive animals. It warrants further investigation of the effect of captopril before the establishment of hypertension and the corresponding alterations in the gut microbiome in order to provide a clearer picture of the underlying mechanism.

## 5. Conclusions

In summary, via the establishment of a DOCA-systemic aging model, we demonstrated that metabolic osteoarthritis is triggered by DOCA-induced hypertension, where significant articular cartilage senescence and subchondral bone loss were observed. Meanwhile, our experiment showed that the anti-hypertensive drug captopril exhibited an anti-senolytic effect on the deteriorated cartilage as well as partial restoration to the subchondral bone. Our results revealed the associations between gut microbiota, hypertension, and joint senescence, suggesting the potential role of the gut flora as a shared mechanism for two common age-related disorders. Moreover, the key genera of gut microbes identified may contribute to the discovery of potential therapeutic targets in the gut microbiome for joint degeneration.

## Figures and Tables

**Figure 1 cells-11-03173-f001:**
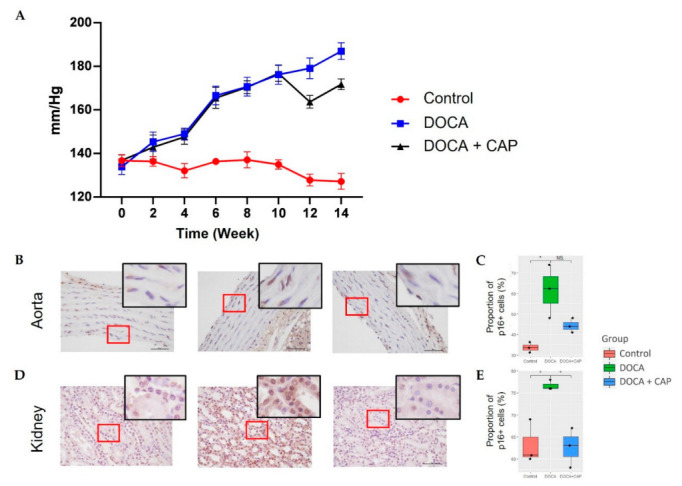
(**A**) Blood pressure measured every 2 weeks from week 0 to week 14 of the Control, DOCA, and DOCA + Captopril groups. The treatment of DOCA starts from w0. In week 9, DOCA + Captopril groups began to feed captopril (*n* = 8 for each group). The results of p16 staining in the aorta (**B**) and kidney (**D**) with red boxes highlighting the p16-positive cells. (**C**,**E**) show the percentage of p16-positive cells in the aorta and kidney, respectively. (* *p* < 0.05).

**Figure 2 cells-11-03173-f002:**
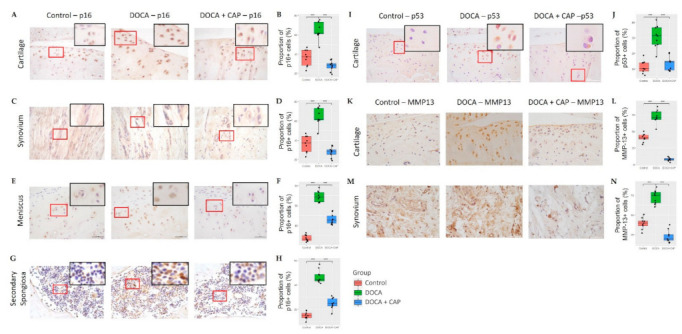
The p16 staining results of cartilage (**A**), synovium (**C**), meniscus (**E**), and secondary spongiosa (**G**) are shown with the red boxes indicating the p16- positive regions. The percentage of p16-positive cells in the (**B**) cartilage, (**D**) synovium, (**F**) meniscus, and (**H**) secondary spongiosa in Control (red), DOCA (green), and Captopril-treated group (blue) are plotted. (**I**,**J**) The results of p53 staining of cartilage and the percentage of p53-positive cells in the sample. The MMP13 staining of the (**K**,**L**) cartilage and (**M**,**N**) synovium of the Control, DOCA, and DOCA + Captopril groups, respectively. (*** *p* < 0.001).

**Figure 3 cells-11-03173-f003:**
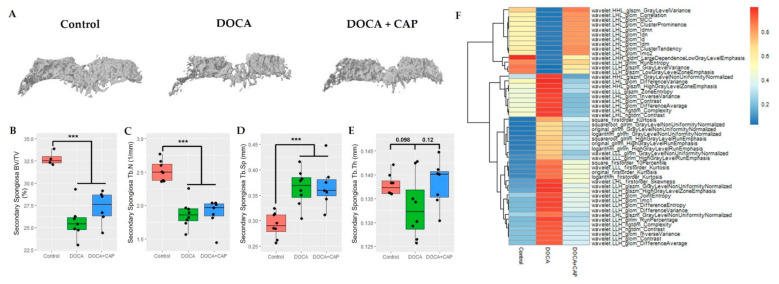
(**A**) The 3D bone structure of secondary spongiosa under micro-CT with the (**B**) bone volume ratio (BV/TV), (**C**) trabecular number (Tb.N), (**D**) trabecular separation (Tb.Sp), and (**E**) trabecular thickness (Tb.Th) calculated for the Control, DOCA, and DOCA+Captopril groups, respectively. (*** *p* < 0.001) (**F**) The heatmap of the mean values of the top 50 radiomic features quantifying the texture of secondary spongiosa.

**Figure 4 cells-11-03173-f004:**
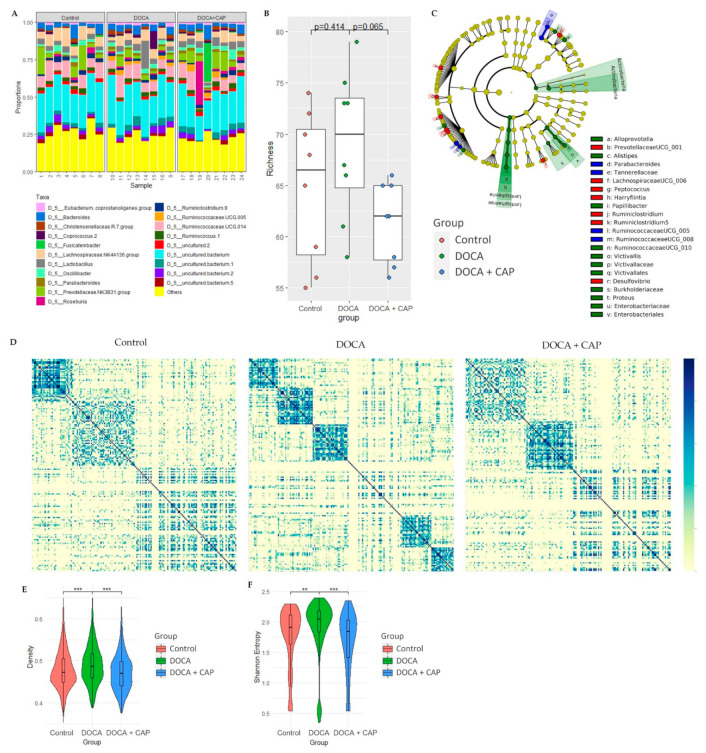
(**A**) Relative abundance TAXA plot of the top 20 most abundant genus-level taxonclassified sequences from gut bacterial 16S rRNA gene amplicon sequencing (*n* = 24). (**B**) The gut microbial genus richness of the Control, DOCA, and DOCA + Captopril groups. A Mann–Whitney U test was conducted, showing a statistically insignificant (*p* = 0.414) difference between the richness of the control and DOCA groups, while a marginally significant difference could be observed among the DOCA and DOCA + Captopril groups (*p* = 0.065). (**C**) Cladogram to demonstrate the linear discriminant effect size of the significant genera. (**D**) Gut microbial genus correlation networks of the Control, DOCA, and DOCA + Captopril groups, respectively. (**E**) Density and (**F**) Shannon entropy of the correlation networks of the Control (red), DOCA (green), and DOCA + Captopril (blue) groups. All the network parameters were compared among the groups through one-way ANOVA with Tukey HSD posthoc analysis (** *p* < 0.01, *** *p* < 0.001).

**Figure 5 cells-11-03173-f005:**
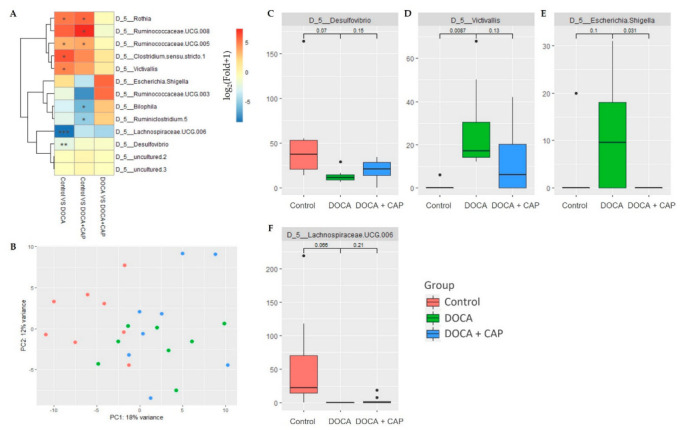
(**A**) Heatmap of the log-fold-change of the top 13 genus having differential abundance among the Control, DOCA, and DOCA + Captopril groups. (**B**) The plot of the first 2 principal components (PCs) of the identified bacterial genera’s abundance, with the first PC explaining 18% variance and the second PC explaining 12% variance. (**C**–**F**) Box plots show the abundance of the 4 highlighted bacterial genera, and the *p*-values of pairwise comparison are reported. (* *p* < 0.05, ** *p* < 0.01, *** *p* < 0.001).

**Figure 6 cells-11-03173-f006:**
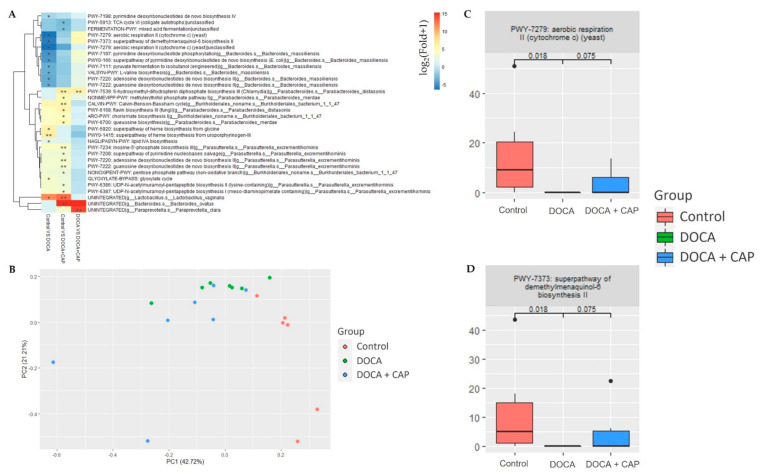
(**A**) Heatmap of the log-fold-change of the top 32 differentially expressed pathways identified from the taxonclassified sequences from gut bacterial shotgun metagenomic sequencing among the Control, DOCA, and DOCA + Captopril groups. (**B**) The plot of the first 2 principal components (PCs) of the 32 identified bacterial pathways, with the first PC explaining 18% variance and the second PC explaining 12% variance. (**C**,**D**) Box plots demonstrate the expression of the 2 identified bacterial pathways in each group with the *p*-values of the group comparison reported. (* *p* < 0.05, ** *p* < 0.01).

**Figure 7 cells-11-03173-f007:**
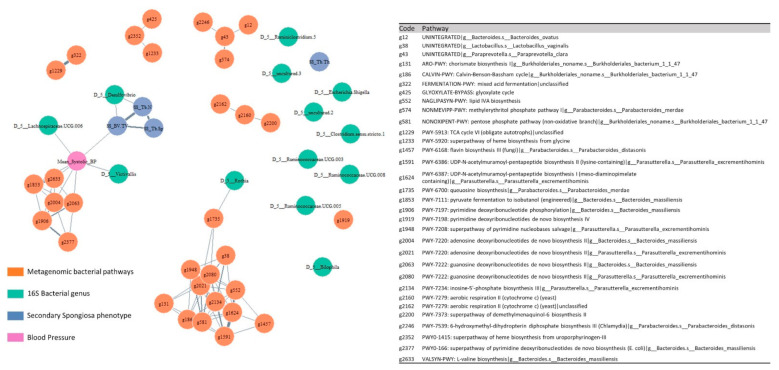
Multi-omics correlation networks between the selected metagenomic bacterial pathways, 16S bacterial genus, phenotypes secondary spongiosa, and blood pressure of all experimental groups. The nodes are colored in orange for the pathways enriched from metagenomic data, cyan for bacterial genus identified from 16S rRNA Gene Sequencing, blue for secondary spongiosa phenotype, and pink for blood pressure. The edges of the networks are defined by Spearman’s correlation coefficients with Benjamini and Yekutieli-adjusted *p*-value < 0.05. Greater intensity and thickness of the edges represent the higher strength of the correlation. An index is shown on the right for the mapping between the enriched bacterial pathways and their corresponding codes.

## Data Availability

Not applicable.

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
