# Peer review of "Captopril Alleviates Chondrocyte Senescence in DOCA-Salt Hypertensive Rats Associated with Gut Microbiome Alteration"

_cells, 2022, doi:10.3390/cells11193173_

Round 1
Reviewer 1 Report
Chan and collaborators investigated the protective role of captopril in OA linked to hypertension, and found this drug alleviates cellular senescence associated with alterations in the gut microbiota especially the genus Escherichia and Shigella. The manuscript is interesting and contributes to novel insights to treating ageing-related diseases.
Major points:
1) The manuscript needs extensive english revision. Sometimes sentences are also incomplete and hard to be understood.
2) The main criticisms are: i) the lack of a normotensive group treated with captopril; this would provide evidence on whether the drug could prevent senescence and thus, OA, at doses even lower than those used to treat hypertension. ii) lack of analysis of the individual microbiomas prior to hypertension induction and prior to captopril treatment as this would provide a more complete picture of variations amongst samples.
Author Response
|
Comments |
Author response |
|
The manuscript needs extensive english revision. Sometimes sentences are also incomplete and hard to be understood. |
We thank the reviewer for pointing it out, a more thorough revision has been made to correct the incomplete or grammatically erroneous sentences. |
|
Lack of a normotensive group treated with captopril; this would provide evidence on whether the drug could prevent senescence and thus, OA, at doses even lower than those used to treat hypertension. |
We well appreciate the reviewer’s comment; however, we would like to clarify that the major focus of our study is to evaluate the treatment effect rather than the preventive effect of captopril on restoring joint deterioration given rise by hypertension. |
Reviewer 2 Report
This is a very interesting study designed to look at the effects of captopril (i.e. antihypertensive medication) on gut microbiota and therapeutic benefits in osteoarthritis. The authors are commended on the study design and the clear presentation of the results. Below please find a few suggestions that in my opinion would strengthen the study:
-The power analysis description is unclear: which variable was used to conduct the power analysis? In other words, when you say the pooled standard deviation of 44.4897 unit, what variable are you referring to?
-The control animals didn't receive salt in their water and are thus different then the DOCA animals. Could you elaborate on the potential influence (if any) of the lack of dietary sodium in the controls and how that could affect the microbiome and/or RAAS function?
- Could you elaborate on the rationale for starting captopril only at week 9? ..as opposed to from the beginning of DOCA administration?
-Line 547: the statement that captopril restored the microbial structure in the gut may be overstated. Microbiome richness was not significant (p = 0.065 indicated the lack of significance which has been set to 0.05 by the power analysis). Perhaps please consider re-stating these results.
-Are you anticipating that any antihypertensive medication would have similar effects? In other words, is it the hypertension that would evoke the mechanisms in osteoarthritis and gut disbiosis?
-You refer to captopril as hypotensive medication, or the animals receiving it in a hypotensive state. Please note that hypotension means abnormally low BP, which neither group of animals in your study show to have. Rather this group of medications (that lowers BP) is referred to as anti-hypertensive medications.
Author Response
|
Comments |
Author’s response |
Author’s Actions |
|
The power analysis description is unclear: which variable was used to conduct the power analysis? In other words, when you say the pooled standard deviation of 44.4897 units, what variable are you referring to? |
We thank the reviewer for pointing out the unclarity regarding the power analysis and justification for sample size in the experiment. We referred to the mean and standard deviation of blood pressure data of the control and DOCA groups reported by Bae et al.[1] (Table 2). Then using the online statistical tool[2], we calculated the required sample size to be at least 5 per group in order to achieve a power of 80% and a significant level of 5%. To ensure achieving higher statistical power, we further increased the sample size to 8 per group.
Reference: 1. Bae, E.H.; Kim, I.J.; Ma, S.K.; Kim, S.W. Rosiglitazone prevents the progression of renal injury in DOCA-salt hypertensive rats. Hypertension Research 2010, 33, 255-262. 2. Dhand, N. K., & Khatkar, M. S. (2014). Statulator: An online statistical calculator. Sample Size Calculator for Comparing Two Independent Means. Accessed 28 September 2022 at http://statulator.com/SampleSize/ss2M.html
|
For a higher level of clarity in the sample size estimation part, we removed the confusing pooled standard deviation in line 70 and refined the description for this section in lines 70-77. |
|
The control animals didn't receive salt in their water and are thus different than the DOCA animals. Could you elaborate on the potential influence (if any) of the lack of dietary sodium in the controls and how that could affect the microbiome and/or RAAS function? |
The diet fed to all rats contains sodium. The rat feed only needs 0.05% sodium, which can meet the daily sodium demand of rats. The sodium content in the feed can meet the daily demand. |
|
|
Could you elaborate on the rationale for starting captopril only at week 9? As opposed to from the beginning of DOCA administration? |
In our experiment, instead of focusing on preventive effect of captopril on joint deterioration, we paid more attention to the treatment effect of the drug. Therefore, we commence the captopril administration at week 9 to ensure the rise in blood pressure stabilizes in the samples which in turn better demonstrates the anti-hypertensive effect and hence any restoration/reversion of the deteriorated joint. |
|
|
Line 247: the statement that captopril restored the microbial structure in the gut may be overstated. Microbiome richness was not significant (p = 0.065 indicated the lack of significance which has been set to 0.05 by the power analysis). Perhaps please consider re-stating these results. |
The reviewer’s comment is well taken, we believe our result might only demonstrate a trend instead of an established restoration effect. We have amended the over-statement in the result part. |
We have changed the statement into a more relaxed version “Furthermore, it significantly changed the organization of gut microbiome in terms of density and Shannon entropy back to the level similar to the control. Therefore, it is evident that captopril might show a trend to partially restore the certain microbial macroscopic structure in the gut.” In lines 249-252 with a less definitive tone on the result. Specifically, given the insignificant finding of the microbiome richness measure, we now only mention captopril might have a partial effect on the microbial structure. |
|
Are you anticipating that any antihypertensive medication would have similar effects? In other words, is it hypertension that would evoke the mechanisms in osteoarthritis and gut dysbiosis? |
We haven't tried other antihypertensive drugs for the time being, but we have used spontaneously hypertensive rats to verify again. Compared with the normal blood pressure rats in the control group, the joints of spontaneously hypertensive rats also show signs of senescence. After hypotension with Captopril, senescence decreased, which was consistent with the condition of hypertensive rats induced by drugs. |
|
|
You refer to captopril as hypotensive medication, or the animals receiving it in a hypotensive state. Please note that hypotension means abnormally low BP, which neither group of animals in your study show to have. Rather this group of medications (that lowers BP) is referred to as anti-hypertensive medications. |
Thank you for your correction, we have changed the wordings of “hypotensive” to “anti-hypertensive” in the description of captopril’s effect. |
“Hypotensive” has been changed to “anti-hypertensive” in lines 53, 63, 322, and 397. |
Round 2
Reviewer 1 Report
The manuscript has improved in terms of grammar, but still needs moderate english revision.
In the first round of revisions, the importance of addressing the effects of captopril in the joint of normotensive animals was suggested. The authors replied that their objective was to analyse the therapeutic rather than the preventive effects of the drug. Nonetheless, it is important for the authors to discuss captopril potential to prevent joint damage. They also must make clear in the introduction section their main objective..
